# Spatial Distribution and Temporal Dynamics of *Candidatus* Liberibacter Asiaticus in Different Stages of Embryos, Nymphs and Adults of *Diaphorina citri*

**DOI:** 10.3390/ijms24108997

**Published:** 2023-05-19

**Authors:** Xiaoge Nian, Shujie Wu, Jiayun Li, Yaru Luo, Jielan He, Shijian Tan, Desen Wang, Yijing Cen, Yurong He

**Affiliations:** 1Department of Entomology, College of Plant Protection, South China Agricultural University, Guangzhou 510642, China; nianxiaoge0629@126.com (X.N.);; 2Henry Fok School of Biology and Agriculture, Shaoguan University, Shaoguan 512005, China

**Keywords:** *C*Las, *D. citri*, distribution, dynamic

## Abstract

Huanglongbing, a globally devastating citrus disease, is associated with *Candidatus* Liberibacter asiaticus (*C*Las) and is mainly transmitted by *Diaphorina citri*. Verification of the distribution and dynamics of *C*Las in *D. citri* is critical to understanding *C*Las transmitted by vectors in nature. Here, the distribution and titers of *C*Las in different sexes and tissues of *D. citri* adults were investigated by fluorescence in-situ hybridization (FISH) and quantitative real-time PCR (qRT-PCR). Results showed that *C*Las had widespread distribution in the brain, salivary glands, digestive system, and reproductive system of both females and males, indicating a systemic infection of *C*Las in *D. citri*. Moreover, *C*Las fluorescence intensity and titers were significantly increased in both the digestive system and the female reproductive system with development and there was a marked decreased in both the salivary glands and the male brain, but there was no significant change in the female brain or the male reproductive system. Furthermore, the distribution and dynamics of *C*Las in embryos and nymphs were investigated. *C*Las was observed in all laid eggs and subsequent first–second-instar nymphs, indicating that a high percentage of embryos and nymphs resulting from infected *D. citri* mothers were infected with CLas.

## 1. Introduction

Huanglongbing (HLB), also called citrus greening disease, is the most fatal disease in citrus varieties worldwide, which results in fruit drop, reduced fruit quality, and twig-and-branch dieback [1,2]. HLB is associated with three known Gram-negative, phloem-limited bacteria: *Candidatus* Liberibacter africanus (*C*Laf), *Candidatus* L. asiaticus (*C*Las), and *Candidatus* L. americanus (*C*Lam). Among them, *C*Las is the most prevalent pathogen and is transmitted by *D. citri* Kuwayama (Hemiptera: Psyllidae) [3], causing substantial economic losses in citrus-growing regions in Asia, North America, and Brazil in South America [1]. Currently, controlling the insect vector is the most critical step for HLB prevention and management since *C*Las is non-culturable in vitro.

*C*Las is transmitted by *D. citri* in a persistent and propagative manner to complete its horizontal transmission from *C*Las-positive plants to healthy plants [4]. After ingestion of *C*Las-positive sap by psyllids, *C*Las first colonizes and propagates in the digestive tract, then moves through the haemocoel, and eventually reaches the salivary glands of *D. citri*, where it is inoculated into a new plant during feeding [5,6]. During the circulative journey, *C*Las breaks multiple barriers, such as the midgut-infection barrier, dissemination barrier, and salivary-escape barrier. Using fluorescence in-situ hybridization (FISH) or quantitative polymerase chain reaction (qPCR), several investigations have been conducted to understand *C*Las colonizing in most *D. citri* tissues, such as the filter chamber, midgut, salivary glands, fat body, ovaries, etc. [5,7,8,9,10]. More recently, Chanim et al. [11] used FISH and immunogold labeling to describe *C*Las accumulation in the cells of psyllid midgut. *C*Las was observed to be in abundance in tubular and quasispherical forms in various psyllid organs by transmission electron microscopy and immunogold labeling [9]. However, previous studies mainly focused on the detection of *C*Las at the early stage of *D. citri*. Little is known about the distribution and titers of *C*Las in different sexes and tissues of *D. citri* adults with development.

Transovarial transmission of microbes such as fungi, bacteria, and viruses from mother to offspring is a common phenomenon in nature that has great significance to maintaining a source of infection and its epidemiology [12,13,14,15]. Depending on the transmission mode, microbes are classed into four categories: non-persistent, semi-persistent, circulative–non-propagative, and circulative–propagative. So far, all circulative–propagative viruses, such as reovirus and tenuivirus, and a few circulative–nonpropagative viruses, such as begomoviruses, have been confirmed to transovarially transmit [15,16]. *C*Las is a typical propagative–circulative bacteria, and a low rate of transovarial transmission by *D. citri* was shown in previous studies [17,18]. As only 3.6–18% of offspring derived from *C*Las-positive adults were identified by qPCR, transovarial transmission appears to contribute very little to *C*Las transmission [17,18]. However, there is no research about the distribution and dynamics of *C*Las in eggs and, subsequently, nymphs derived from females reared on *C*Las-positive plants.

In the current study, we combined FISH and quantitative real-time PCR (qRT-PCR) technology to investigate the distribution and titers of CLas in different stages of embryos, nymphs, and adults of *D. citri*.

## 2. Results

### 2.1. Distribution and Relative Titers of CLas in CLas-Positive Adult Males

The distribution and relative titers of *C*Las in *C*Las-positive adult males were investigated by FISH and RT-PCR. FISH results showed that *C*Las was detected in the brain and salivary glands of adult males at 5 days after eclosion (5 DAE), 10 DAE, and 15 DAE, and was not found in the brain or salivary glands of *C*Las-negative males at 15 DAE (Figure 1A). *C*Las fluorescence intensity was significantly decreased in the brain (*F*_2,11_ = 24.93; *p* < 0.001) and salivary glands (*F*_2,11_ = 18.24; *p* < 0.001) with development (Figure 1B,C). Moreover, the relative *C*Las titers showed a gradual decreasing trend in both the brain (*F*_2,11_ = 5.54; *p* < 0.05) and the salivary glands (*F*_2,11_ = 49.24; *p* < 0.001) (Figure 1D,E).

In addition, *C*Las had a widespread distribution in the male gut and Malphigian tubules (Figure 2A). *C*Las was not detected in the gut or Malphigian tubules of *C*Las-negative males at 15 DAE (Figure 2A). Along with the development of males, *C*Las fluorescence intensity was significantly increased in the Malphigian tubules (*F*_2,11_ = 14.89; *p* < 0.01), but there was no significant change in the gut (*F*_2,11_ = 3.22; *p* > 0.05) (Figure 2B,C). Moreover, the relative *C*Las titer in the digestive system obviously increased with development (*F*_2,11_ = 99.86; *p* < 0.001) (Figure 2D).

In Figure 3A, *C*Las was detected in the male reproductive system, including the testis and sperm sac. However, no *C*Las signal was detected in the testis and sperm sac of *C*Las-negative males at 15 DAE. For *C*Las fluorescence intensity in the male testis, there was no significant difference during development (*F*_2,11_ = 4.19; *p* > 0.05) (Figure 3B), but the fluorescence intensity of *C*Las in sperm sacs at 15 DAE was obviously weaker than those at 5 DAE and 10 DAE (*F*_2,11_ = 6.76; *p* < 0.05) (Figure 3C). As for the relative *C*Las titer, there was no significant difference in the male reproductive system with development (*F*_2,11_ = 4.01; *p* > 0.05) (Figure 3D).

Taken together, *C*Las fluorescence intensity was highest in the Malpighian tubules, followed by the gut, brain, and sperm sac, and lowest in the salivary glands and testes of adult males (5 day-old males: *F*_5,23_ = 94.53, *p* < 0.001; 10 day-old males: *F*_5,23_ = 122.4, *p* < 0.001; 15 day-old males: *F*_5,23_ = 199.9, *p* < 0.001) (Appendix A). The relative *C*Las titer was highest in the digestive system (5 day-old males: *F*_3,15_ = 35.53, *p* < 0.001; 10 day-old males: *F*_3,15_ = 89.94, *p* < 0.001; 15 day-old males: *F*_3,15_ = 404.4, *p* < 0.001) (Appendix A).

### 2.2. Distribution and Relative Titers of CLas in CLas-Positive Adult Females

As shown in Figure 4A, *C*Las had widespread distribution in the *C*Las-positive adult female brain and salivary glands. There was no *C*Las signal in the *C*Las-negative adult female brain or salivary glands. In addition, fluorescence intensity (*F*_2,11_ = 0.74; *p* > 0.05) and the relative titer (*F*_2,11_ = 0.07; *p* > 0.05) of *C*Las in the female brain showed no significant difference during different developmental stages (Figure 4B,D), but those in the female salivary glands displayed a significant decrease at 10 DAE and 15 DAE compared with 5 DAE (*F*_2,11_ = 12.03; *p* < 0.01; *F*_2,11_ = 22.50; *p* < 0.001) (Figure 4C,E).

*C*Las was also found to localize in the gut and Malphigian tubules of *C*Las-positive females (Figure 5A). In addition, *C*Las was not detected in the gut or Malphigian tubules of *C*Las-negative females at 15 DAE (Figure 5A). There was no obvious difference in *C*Las fluorescence intensity in the female gut (*F*_2,11_ = 0.02; *p* > 0.05) (Figure 5B) but was significantly increased in the Malphigian tubules during development (*F*_2,11_ = 37.67; *p* < 0.001) (Figure 5C). In addition, the relative *C*Las titer in the female digestive system increased markedly with development (*F*_2,11_ = 31.67; *p* < 0.001) (Figure 5D).

*C*Las was also detected in the female reproductive system, which includes the ovaries and spermathecae, but was not detected in the ovaries or spermathecae of *C*Las-negative females at 15 DAE (Figure 6A). Most of the ovaries at 5 DAE were small and all oocytes were at the pre-vitellogenic stage, but at 10 DAE they contained both pre-vitellogenic and vitellogenic oocytes, and the mature oocytes of elder females (15 DAE) were filled with *C*Las. *C*Las fluorescence intensity in the follicles increased significantly along with ovarian development (*F*_2,11_ = 63.13; *p* < 0.001), eventually filling with mature follicles, but the fluorescence intensity of *C*Las in the spermathecae decreased significantly (*F*_2,11_ = 29.91; *p* < 0.001) (Figure 6B,C). The relative *C*Las titer in the female reproductive system significantly increased with development (*F*_2,11_ = 65.19; *p* < 0.001) (Figure 6D).

In summary, *C*Las fluorescence intensity was highest in female Malpighian tubules, followed by the gut, brain, and spermathecae, and lowest in the ovaries of *C*Las-positive females at 5 DAE (*F*_5,23_ = 118.2, *p* < 0.001). In *C*Las-positive females at 15 DAE, *C*Las density was highest in the Malpighian tubules, followed by the gut, brain, and ovaries, and lowest in the spermathecae and salivary glands (*F*_5,23_ = 144.3, *p* < 0.001) (Appendix A). The relative *C*Las titer was highest in the female digestive system, followed by the female reproductive system and brain, and lowest in the salivary glands (5 day-old males: *F*_3,15_ = 102.7, *p* < 0.001; 10 day-old males: *F*_3,15_ = 77.24, *p* < 0.001; 15 day-old males: *F*_3,15_ = 280.6, *p* < 0.001) (Appendix A).

### 2.3. The Distribution and Dynamics of CLas in Embryos and Nymphs

In embryos observed at 18–24 h post-oviposition, the formed germ band extended and the proto-bacteriome appeared to be cellularized. At 42–48 h post-oviposition, the embryos were in the process of dorsal closure. The former cells were fused into a syncytium at 66–72 h post-oviposition, and then the embryos hatched at 90–96 h post-oviposition. *C*Las was observed throughout embryo development and the subsequent first–second-instar nymphs obtained from CLas-positive females (Figure 7A). No *C*Las signal was detected in the eggs or subsequent first–second-instar nymphs derived from *C*Las-negative females (Figure 7B). There was no significant difference in *C*Las fluorescence intensity during embryogenesis and first-instar nymphs, but it significantly increased in second-instar nymphs (*F*_2,11_ = 12.33; *p* < 0.001) (Figure 7C). Moreover, the relative *C*Las titer showed no significant difference during embryogenesis but significantly increased in the subsequent first–second-instar nymphs (*F*_2,11_ = 24.55; *p* < 0.001) (Figure 7D).

## 3. Discussion

Confocal imaging using FISH with a *C*Las probe enables us to visualize *C*Las’s location within vector insects. In the present study, the near-systemic invasion of *C*Las within all female and male dissected tissues, including the brain, salivary glands, gut, Malpighian tubules, and reproductive system, suggested a wide distribution of *C*Las in *C*Las-positive psyllids, which is in accordance with previous studies [4,7,8,9,10,19,20,21]. It was obvious that the *C*Las titer in certain tissues of *D. citri* was very low. Whether via relative quantification using normalization to a reference gene or absolute quantification using a standard curve, errors are inevitable in detection. FISH technology is visual and sensitive, and even weak probe signals can be captured. Therefore, by combining the results of FISH and qRT-PCR, the relative differences in *C*Las titers in *D. citri* can be quantified in various tissues at different ages/stages of the vector.

Along with adult development, the trend of *C*Las fluorescence intensity by FISH was consistent with that of the *C*Las titer by qRT-PCR in the same tissue. The *C*Las titer was highest in the digestive system in our study, which is consistent with previous reports [7,8,10]. However, *C*Las fluorescence intensity by FISH was highest in the Malpighian tubes in all examined tissues. These differing results may, on the one hand, be due to the difference in detection techniques. The FISH technique could be used to quantify and visualize the difference in *C*Las in any tissue of *D. citri*. For example, it could separately quantify *C*Las fluorescence intensity in the gut and Malpighian tubules, in female ovaries and the spermathecae, or in the male testis and sperm sac. However, qRT-PCR could only be used to quantify the corresponding digestive and reproductive systems because these tissues are closely connected in *D. citri*.

*C*Las fluorescence intensity and titers were significantly increased in both the digestive system and the female reproductive system with development, showed a marked decreased in both the salivary glands and the male brain, and showed no significant change in the female brain or male reproductive system. Ammar et al. [22] described the detailed ultrastructure of various tissues, and the epithelial cells of Malpighian tubules were vacuolated and lined with extensive microvilli in *D. citri*. Malpighian tubules in insects have discrete compartments and specific cell types that are highly specialized in excretory function [23]. In addition to *C*Las, the *Wolbachia* titer was highest in the Malpighian tubules [10]. The cause of these densely aggregated microbes in the Malpighian tubules is not yet known. It is possible that nitrogen metabolites in Malpighian tubules are essential for the development and replication of microbes. The effect of a large number of *C*Las gathered in the Malpighian tubes on the interactions between *D. citri* and *C*Las requires further study. The *C*Las fluorescence intensity and titer were lowest in the salivary glands of both males and females, which is consistent with a previous report [7]. Furthermore, the fluorescence intensity and titer of *C*Las were lower and gradually decreased in the sperm sac as it developed, which provided a possible explanation for the low sexual-transmission efficiency (3.6%) of *C*Las from *C*Las-positive males to *C*Las-negative females during mating [24].

Previous studies reported that *C*Las was detected in the ovaries at a low frequency by qRCR [7,8,10]. In addition, Hosseinzadeh et al. (2019) reported that *C*Las was occasionally found in the follicle cells of females [25]. In the current study, *C*Las was detected in all female reproductive systems by FISH. At the early stage of ovarian development, there was no *C*Las dissected in the follicles by FISH, and *C*Las fluorescence intensity in the follicles increased significantly along with ovarian development, eventually filling with mature follicles. The corresponding *C*Las titer in the female reproductive system was significantly increased by RT-PCR. The differing results between these studies may have arisen from different developmental stages of *D. citri.* Previously published studies used psyllids at a certain stage, whereas we used insect vectors at three different developmental stages.

Transovarial transmission of microbes from female vectors to progeny is an important strategy for microbes to speed up transmission and is recognized as crucial to becoming an epidemic [13,14]. Transovarial transmission controls microbe dispersal in space and time and thus plays an important role in microbe outbreak, which has received constant attention [24,26,27]. However, most studies have focused on circulative–propagative or circulative–non-propagative plant viruses [15,28,29,30], and only a few have investigated circulative–propagative plant bacteria [24,26]. *C*Las moves through the psyllid body in a propagative–circulative manner [7,8]. Until now, only two cases have reported that *C*Las can be vertically transmitted at a low rate [17,18], which is contrary to our results.

Kelley and Pelz-Stelinski (2019) reported that nymphs exposed to *C*Las were more likely to pick it up via horizontal transmission than vertical transmission [18]. In order to reduce this possibility, partially CLas-resistant citrus plants—orange Jasmine (*Murraya panifulata* (L.) Jack) [31]—were used for egg laying by infected females to study transovarial transmission. *C*Las was observed in all laid eggs and subsequent first–second-instar nymphs, indicating that a high percentage of embryos and nymphs resulting from the infected *D. citri* mothers were infected with *C*Las. Although in *M. panifulata C*Las decreased gradually to an undetectable level, it is also possible that the older nymphs may have acquired the pathogen from the oviposition site. In addition, the epidermis of the nymph thickens gradually along with the development of the nymph, which makes it difficult for the nymphs to sufficiently decolorize, so the *C*Las probe could not penetrate the cuticle in the third–fifth-instar nymphs. Hence, the distribution and relative titers of *C*Las in subsequent third–fifth-instar nymphs were not shown in the present work. It is well documented that even with highly infected colonies of *D. citri*, when tested individually they transmit this bacterium to only up to 20–46% of test plants in different regions around the world [32]. As for transovarial transmission of *C*Las in *D. citri*, much lower percentages have been reported [17]. In future studies, actual transovarial transmission of *C*Las from infected *D. citri* mothers to their offspring needs to be verified by feeding first-generation nymphs, individually or in small groups, on young, susceptible, healthy citrus plants. These tests could show whether the high percentage of transovarially infected nymphs would lead to efficient inoculation of *C*Las into these test plants and the potential significance of transovarial transmission in the epidemiology of *C*Las/HLB.

*C*Las, one of the world’s most serious citrus pathogens, is considered to have spread from China to the rest of the world around the late 1800s [33]. *D. citri* invasion and the movement of *C*Las-infected materials are recognized as two important factors that are attributed to the rapid spread of *C*Las around the world. In the current study, *D. citri* efficiently transovarially transmitted *C*Las to their offspring. *C*Las-positive eggs—which are so small as to be almost invisible—laid by *C*Las-infected psyllids on plants can be transported over long distances with plant material by human activity. Thus, transovarial transmission may also make a significant contribution to the global spread of *C*Las in addition to the outbreak of HLB in fields.

## 4. Materials and Methods

### 4.1. Host Plants and Insect Colonies

The CLas-negative (healthy) *D. citri* colony was obtained from a laboratory culture continuously reared on healthy *Citrus limon* (L.) Osbeck plants. *C*Las-positive *D. citri* were obtained from a subculture of CLas-negative *D. citri* reared on *C*Las-positive *C. limon*. The two colonies were kept in the different incubators at 26 ± 1 °C, 65 ± 5% relative humidity (RH) with a 14L: 10D photoperiod. Both healthy and *C*Las-positive cultures were periodically tested to confirm the absence or presence of CLas by qPCR as previously reported [34]. The mean infection rate of *C*Las in the *D. citri* reared on *C*Las-positive plants was 90–100%. In order to explore the dynamics of transovarially transmitted *C*Las in *D. citri,* healthy orange Jasmine (*M. panifulata* (L.) Jack) plants (relatively resistant to CLas) were used for egg laying by infected females to study the dynamics of *C*Las in different stages of embryos and subsequent nymphs of *D. citri*.

### 4.2. Tissue Dissection and Sample Collection

Newly emerged adults (within 24 h) reared on *C*Las-positive plants were collected and relocated onto new *C*Las-positive plants, which were marked as 1 day after eclosion (1 DAE). Three different developmental stages (5 DAE, 10 DAE, and 15 DAE) of adult females and males were prepared. In the laboratory, adults were first anesthetized with CO_2_, the brain was connected to the salivary glands and the gut to the Malpighian tubules, and the reproductive system was then excised in 1 × phosphate-buffered saline (PBS) (Sangon Biotech, Shanghai, China) under a dissecting microscope (Motic SMZ-171). Following dissection, each sample was divided into two parts for conjoint analysis, one for FISH to detect *C*Las fluorescence intensity and the other for RT-PCR for relative *C*Las titers. All experiments were performed with four biological replications. Each replication included 40–45 insects for qRT-PCR. Four FISH tests were conducted and 60 samples per tissue were imaged to confirm the repeatability. In these experiments, the *D. citri* reared on *C*Las-positive plants had a high titer with a cycle threshold (CT) of less than 25 in qPCR monitoring and a 100% infection rate by FISH. The adult females and males of *D. citri* reared on *C*Las-negative *C. limon* plants at 15 DAE were dissected for negative control.

### 4.3. Preparation of Embryos and Subsequent Nymphs

In order to explore the dynamics of *C*Las in different stages of embryos and subsequent nymphs of *D. citri,* the female adults reared on the *C*Las-positive plants were allowed to lay eggs on healthy orange Jasmine (*M. panifulata* (L.) Jack) plants. After spawning for 12 h, the females were removed and tested to confirm the presence of *C*Las by qPCR. Results show that the infection rate of *C*Las in females was 100% and that all the females used were *C*Las-positive. According to the morphological characteristics [35,36], embryos of 24 h, 48 h, 72 h, and 96 h post-oviposition and subsequent first (1st)- and second (2nd—instar nymphs were collected. Four biological repetitions were conducted and imaged to confirm repeatability. As for FISH, a total of 200 embryos of each stage and 60 nymphs of each instar were tested. As for qRT-PCR, samples of different stages of embryos and nymphs were gently scraped off the plants with a soft brush, and each repetition included 40–50 nymphs.

### 4.4. Fluorescence In-Situ Hybridization (FISH)

FISH protocol was performed as previously described [8,10,37] with minor modifications. The samples of embryos, nymphs, and dissected adult tissues were fixed in Carnoy’s fixative (glacial acetic acid: ethanol: chloroform, 1:3:6, *v*:*v*:*v*) overnight at room temperature. After washing with 50% ethanol three times (10 min each time), the fixed samples were treated with 6% H_2_O_2_ in 80% ethanol until sufficiently decolorized. Then, the bleached samples were rinsed with PBST (1 × PBS: TritonX-100, 99.7:0.3, *v*:*v*) three times (20 min each time) and pre-incubated three times (10 min per incubation) with the hybridization buffer (20 mM Tris-HCl, pH 8.0, 0.9 M NaCl, 0.01% (wt/wol) sodium dodecyl sulfate, 30% (*v*:*v*) formamide) without the probe.

*C*Las probe (5′-Cy3-CATTATCTTCTCCGGCG -3′) [10] was labeled in 5′-terminal with fluorescence labeling of Cy3. The samples were hybridized within a hybridization buffer containing 10 pmol mL^−1^ of *C*Las probe at 25 °C overnight. After washing three times with PBST (20 min each time), nuclei were stained with DAPI (40, 60-diamidino-2-phenylindole; 0.1 mg mL^−1^) for 15 min and rinsed with PBST for 10 min. The stained samples were transferred to glass slides with spacers and mounted in mounting medium using a cover slip. The slides were viewed under a Leica TCS-SP8 (Leica Microsystems, Exton, PA, USA) confocal microscope using two excitation lasers, exciting near UV-diode 405 nm and 550 nm lasers to detect DAPI and the Cy3-signal, respectively. The acquisition setting and scanning setting were kept fixed throughout all experiments. Additionally, sequential scanning limited the potential signal overlap of the fluorescent probe. Image processing was completed using Leica LAS-AF software (v2.6.0). For the specificity of the probe signal, healthy insects (*C*Las-negative) at 15 DAE and were confirmed as negative controls.

### 4.5. qRT-PCR

The *C*Las primers (:TCGAGCGCGTATGCAATACG; R: GCGTTATCCCGTAGAAAAAGGTAG) used in the study were the same as those previously reported [34]. For relative *C*Las-titer detection, total RNA was extracted using TRIzol reagent (Invitrogen, Carlsbad, CA, USA) The first-strand cDNA was synthesized using the PrimeScript™ II 1st Strand cDNA Synthesis Kit (Takara, Beijing, China) according to the manufacturer’s protocol. Quantitative RT-PCR was performed using TB Green^®^ Premix Ex Taq™ II (Takara, Beijing, China) on an ABI PRISM^®^ 7500 Real-Time System (Applied Biosystems, Foster City, CA, USA). Beta-actin (Dcβ-ACT) was selected to normalize the *C*Las-expression level in *D. citri*.

### 4.6. Statistical Analysis

For the Cy3 channel of tissue and embryo images, the mean grayscale value of the fluorescence intensity of regions of interest (ROI) was analyzed [38]. All the data should be checked for normality and homogeneity of variance in advance. One-way ANOVA followed by the Tukey’s Honestly Significant Difference (HSD) test were used for multiple comparisons (different letters denoted by *p* < 0.05). All statistical analyses were performed using GraphPad Prism 9.0 software.

## Figures and Tables

**Figure 1 ijms-24-08997-f001:**
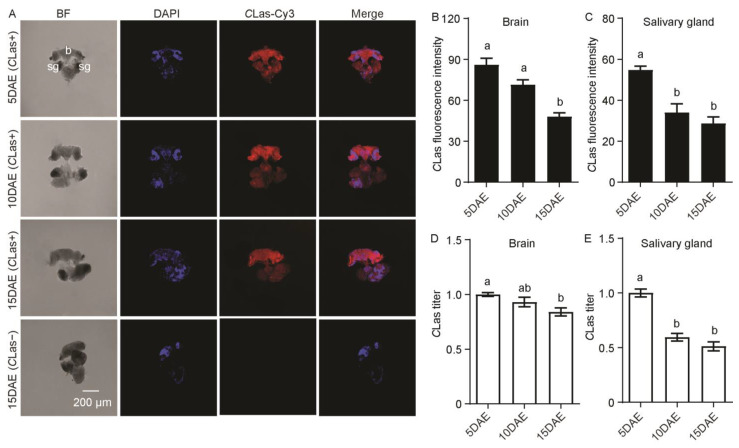
Analysis of distribution and relative titers of *C*Las in the brain and salivary glands of *C*Las-positive adult males by FISH and RT-PCR. (**A**) Representative confocal images of *C*Las distribution in the brain and salivary glands of *C*Las-positive adult males at 5, 10, and 15 DAE and *C*Las-negative adult males at 15 DAE. b: brain, sg: salivary gland. DAE: days after eclosion, *C*Las-: *C*Las-negative, *C*Las+: *C*Las-positive. Scale bar = 200 μm, BF: imaging of bright field allowed for identification of regions in different tissues; DAPI: the cell nuclei was stained with DAPI and visualized in blue. *C*Las-Cy3: The *C*Las signal was marked by Cy3 and visualized in red. Merge: merged imaging of co-localization of cell nuclei and *C*Las. (**B**) *C*Las fluorescence intensity in male brain. (**C**) *C*Las fluorescence intensity in male salivary glands. (**D**) Relative *C*Las titer in male brain. (**E**) Relative *C*Las titer in male salivary glands. The different letters at the top of the columns in (**B**–**E**) figures indicate statistically significant differences (in all cases *p* < 0.05).

**Figure 2 ijms-24-08997-f002:**
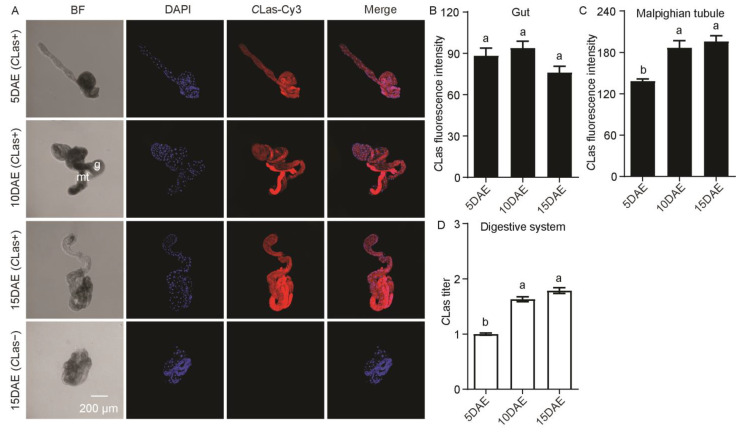
Analysis of distribution and relative titer of *C*Las in *C*Las-positive adult-male digestive system by FISH and RT-PCR. (**A**) Representative confocal images of *C*Las distribution in the digestive system of *C*Las-positive males at 5 DAE, 10 DAE, and 15 DAE and *C*Las-negative adult males at 15 DAE. DAE: days after eclosion. g: gut, mt: Malpighian tubule. *C*Las-: *C*Las-negative, *C*Las+: *C*Las-positive. Scale bar is 200 μm. (**B**) *C*Las fluorescence intensity in male gut. (**C**) *C*Las fluorescence intensity in male Malpighian tubules. (**D**) Relative *C*Las titer in male digestive system. The different letters at the top of the columns in (**B**–**D**) figures indicate statistically significant differences (in all cases *p* < 0.05).

**Figure 3 ijms-24-08997-f003:**
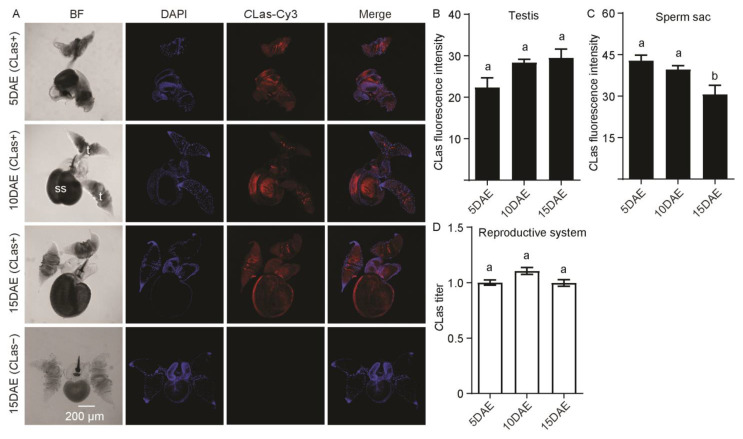
Analysis of distribution and relative titer of *C*Las in *C*Las-positive male reproductive system by FISH and RT-PCR. (**A**) Representative confocal images of *C*Las distribution in the reproductive system of *C*Las-positive males at 5 DAE, 10 DAE, and 15 DAE and *C*Las-negative adult males at 15 DAE. DAE: days after eclosion. t: testis, ss: sperm sac. *C*Las-: *C*Las-negative, *C*Las+: *C*Las-positive. Scale bar is 200 μm. (**B**) *C*Las fluorescence intensity in male testis. (**C**) *C*Las fluorescence intensity in male sperm sac. (**D**) Relative *C*Las titer in male reproductive system. The different letters at the top of the columns in (**B**–**D**) figures indicate statistically significant differences (in all cases *p* < 0.05).

**Figure 4 ijms-24-08997-f004:**
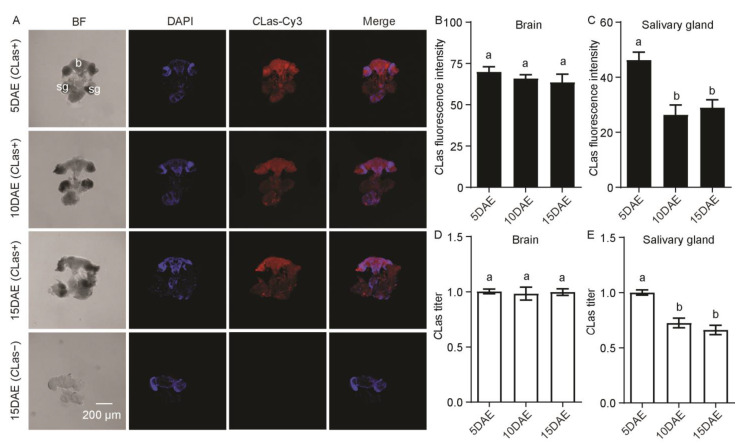
Analysis of distribution and relative titers of *C*Las in *C*Las-positive adult female brain and salivary glands by FISH and RT-PCR. (**A**) Representative confocal images of *C*Las distribution in the brain and salivary glands of *C*Las-positive females at 5 DAE, 10 DAE, and 15 DAE and *C*Las-negative adult females at 15 DAE. DAE: days after eclosion. b: brain, sg: salivary gland. *C*Las−: *C*Las-negative, *C*Las+: *C*Las-positive. Scale bar is 200 μm. (**B**) *C*Las fluorescence intensity in female brain. (**C**) *C*Las fluorescence intensity in female salivary glands. (**D**) Relative *C*Las titer in female brain. (**E**) Relative *C*Las titer in female salivary glands. The different letters at the top of the columns in (**B**–**E**) figures indicate statistically significant differences (in all cases *p* < 0.05).

**Figure 5 ijms-24-08997-f005:**
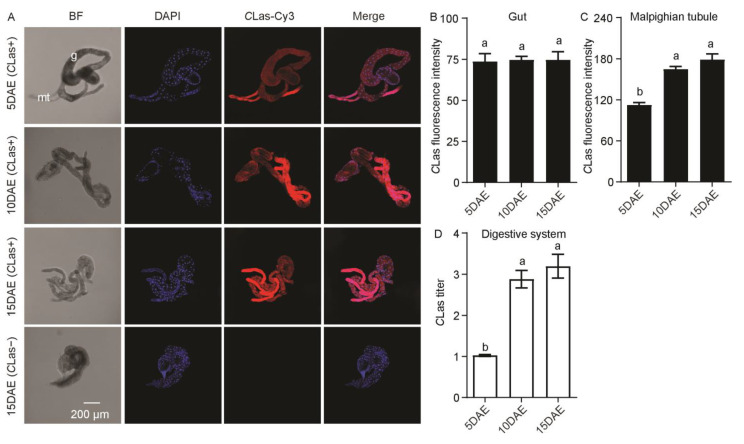
Analysis of distribution and relative titer for *C*Las in *C*Las-positive adult female digestive system by FISH and RT-PCR. (**A**) Representative confocal images of *C*Las distribution in the digestive system of *C*Las-positive females at 5 DAE, 10 DAE, and 15 DAE and *C*Las-negative adult females at 15 DAE. DAE: days after eclosion. mt: Malpighian tubule, g: gut. *C*Las−: *C*Las-negative, *C*Las+: *C*Las-positive. Scale bar is 200 μm. (**B**) *C*Las fluorescence intensity in female gut. (**C**) *C*Las fluorescence intensity in female Malpighian tubules. (**D**) Relative *C*Las titer in female digestive system. The different letters at the top of the columns in (**B**–**D**) figures indicate statistically significant differences (in all cases *p* < 0.05).

**Figure 6 ijms-24-08997-f006:**
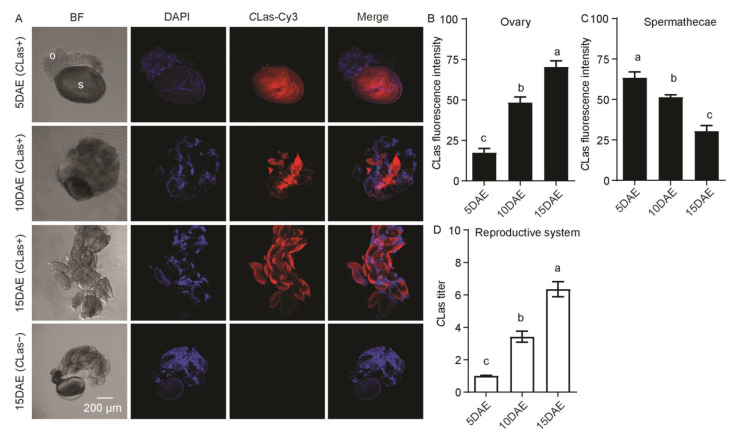
Analysis of distribution and relative titer of *C*Las in *C*Las-positive adult female reproductive system by FISH and RT-PCR. (**A**) Representative confocal images of *C*Las distribution in reproductive system of *C*Las-positive adult females at 5 DAE, 10 DAE, and 15 DAE and *C*Las-negative adult females at 15 DAE. DAE: days after eclosion, o: ovary, s: spermathecae, *C*Las−: *C*Las-negative, *C*Las+: *C*Las-positive, Scale bar is 200 μm. (**B**) *C*Las fluorescence intensity in female ovaries. (**C**) *C*Las fluorescence intensity in female spermathecae. (**D**) Relative *C*Las titer in female reproductive system. The different letters at the top of the columns in (**B**–**D**) figures indicate statistically significant differences (in all cases *p* < 0.05).

**Figure 7 ijms-24-08997-f007:**
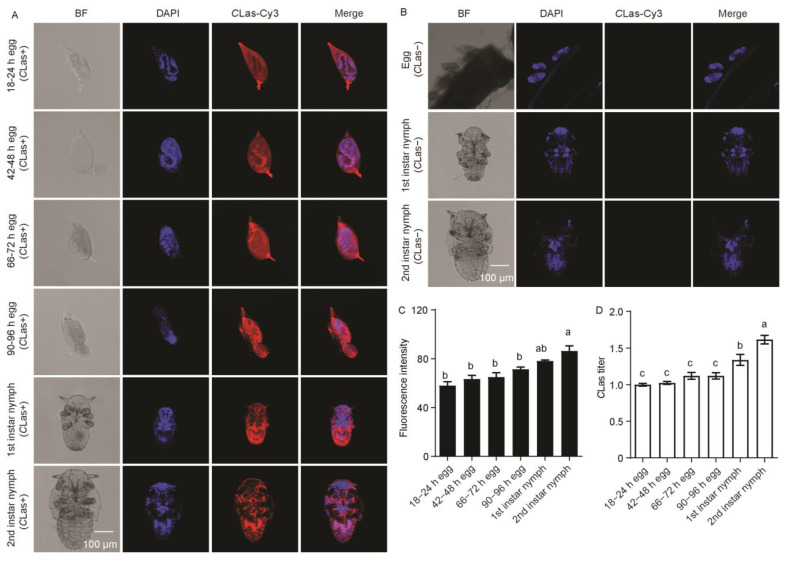
Analysis of distribution and relative titer of *C*Las in the embryos and subsequent 1st–2nd-instar nymphs derived from *C*Las-positive females. (**A**) Representative confocal images of *C*Las distribution during embryogenesis and subsequent 1st–2nd-instar nymphs derived from *C*Las-positive females. Scale bar is 100 μm. (**B**) Representative confocal images of *C*Las distribution in the eggs and subsequent 1st–2nd-instar nymphs derived from *C*Las-negative females. Scale bar is 100 μm. (**C**) *C*Las fluorescence intensity during embryogenesis and subsequent 1st–2nd-instar nymphs. (**D**) Relative *C*Las titer during embryogenesis and subsequent 1st–2nd-instar nymphs. The different letters at the top of the columns in (**C**,**D**) figures indicate statistically significant differences (in all cases *p* < 0.05).

## Data Availability

All the data is in this manuscript.

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
