# Peer review of "Spatial Distribution and Temporal Dynamics of Candidatus Liberibacter Asiaticus in Different Stages of Embryos, Nymphs and Adults of Diaphorina citri"

_ijms, 2023, doi:10.3390/ijms24108997_

Round 1
Reviewer 1 Report (Previous Reviewer 1)
The authors have mostly made the necessary changes required earlier by this reviewer. However, a few suggestions below, to improve accuracy and readability, are considered necessary to make this ms. acceptable for publication:
Line 72. Define DAE before using this short form,
Line 173. After "nymphs" add "obtained from CLas-positive females".
L 197. Add "relatived" before "difference".
L 198. Add "titer" after "CLas".
L 198. Replace "comprehensively and accurately" with " in various tissues at different ages/stages of the vector.
Lines 245-249. Delete the two sentences starting with "Firstly...".
Line 290. Add "transovarially-transmtted" before "CLas in".
Line 290-291. Before "D. citri" delete "different stages of ..etc."
Line 292. After "plants" add "(relatively resistent to CLas)".
Lines 457-458. Refer. No. 32. Replace with " Ammar E-D, Shatters RG, & Heck M. 2020. Huanglongbing pathogens: acquisition, transmission and vector interactions. in Asian citrus psyllid: biology, ecology and management of the Huanglongbing vector (eds, Qureshi JA & Stansly PA.) 113-139 (CAB International, Wallingford).
Minor editing of English language required
Author Response
Dear Editors and Reviewers:
Thank you for your letter and for the reviewers’ comments concerning our manuscript entitled “Spatial distribution and temporal dynamics of Candidatus Liberibacter asiaticus in different stages of embryos, nymphs and adults of Diaphorina citri” (Manuscript ID: ijms-2343073). These comments are all of great important and very helpful for revising and improving our manuscript. We have revised the manuscript carefully according to all your comments. Revised portions are marked with red font in our revised manuscript. Our point-by-point responses to the comments are listed below.
Sincerely,
Xiaoge Nian
Reviewer: 1
The authors have mostly made the necessary changes required earlier by this reviewer. However, a few suggestions below, to improve accuracy and readability, are considered necessary to make this ms. acceptable for publication:
Line 72. Define DAE before using this short form,
Response: Thanks very much for your great suggestion. We have added it in our revised manuscript.
Line 173. After "nymphs" add "obtained from CLas-positive females".
Response: Thanks very much for your great suggestion. We have added it in our revised manuscript.
L 197. Add "relatived" before "difference".
Response: Thanks very much for your great suggestion. We have added it in our revised manuscript.
L 198. Add "titer" after "CLas".
Response: Thanks very much for your great suggestion. We have added it in our revised manuscript.
L 198. Replace "comprehensively and accurately" with " in various tissues at different ages/stages of the vector.
Response: Thanks very much for your great suggestion. We have revised it in our revised manuscript.
Lines 245-249. Delete the two sentences starting with "Firstly...".
Response: Thanks very much for your great suggestion. We have deleted it in our revised manuscript.
Line 290. Add "transovarially-transmtted" before "CLas in".
Response: Thanks very much for your great suggestion. We have added it in our revised manuscript.
Line 290-291. Before "D. citri" delete "different stages of ..etc."
Response: Thanks very much for your great suggestion. We have revised it in our revised manuscript.
Line 292. After "plants" add "(relatively resistent to CLas)".
Response: Thanks very much for your great suggestion. We have added it in our revised manuscript.
Lines 457-458. Refer. No. 32. Replace with " Ammar E-D, Shatters RG, & Heck M. 2020. Huanglongbing pathogens: acquisition, transmission and vector interactions. in Asian citrus psyllid: biology, ecology and management of the Huanglongbing vector (eds, Qureshi JA & Stansly PA.) 113-139 (CAB International, Wallingford).
Response: Thanks very much for your great suggestion. We have revised it in our revised manuscript.
Reviewer 2 Report (New Reviewer)
This manuscript was well written, and the information was quite interesting. I recommend publishing this manuscript. I made several very minor suggestions on an annotated version of the manuscript that I uploaded.
I have one question related to future research. With the pathogen so widespread through various tissues of the host insect, does the pathogen affect the insect’s life-span, fecundity, or behavior?

Author Response
Dear Editors and Reviewers:
Thank you for your letter and for the reviewers’ comments concerning our manuscript entitled “Spatial distribution and temporal dynamics of Candidatus Liberibacter asiaticus in different stages of embryos, nymphs and adults of Diaphorina citri” (Manuscript ID: ijms-2343073). These comments are all of great important and very helpful for revising and improving our manuscript. We have revised the manuscript carefully according to all your comments. Revised portions are marked with red font in our revised manuscript. Our point-by-point responses to the comments are listed below.
Sincerely,
Xiaoge Nian
Reviewer: 2
This manuscript was well written, and the information was quite interesting. I recommend publishing this manuscript. I made several very minor suggestions on an annotated version of the manuscript that I uploaded.
I have one question related to future research. With the pathogen so widespread through various tissues of the host insect, does the pathogen affect the insect’s life-span, fecundity, or behavior?
Response: Thanks very much for your great suggestion. The pathogen can affect the insect’s life-span, fecundity, or behavior. We focus on the effects of the pathogen on the reproduction now.
Line 45. FISH. I know you defined this in the Abstract, but you should probably also define it here. Many readers don't read Abstracts.
Response: Thanks very much for your great suggestion. We have added it in our revised manuscript.
Line 67. qRT-PCR. You should define this the first time it is mentioned in the text, not just in the Abstract.
Response: Thanks very much for your great suggestion. We have added it in our revised manuscript.
Line 122. “these” should change to “those”
Response: Thanks very much for your great suggestion. We have revised it in our revised manuscript.
Line 145. “and not been” should change to “but was not”
Response: Thanks very much for your great suggestion. We have revised it in our revised manuscript.
Line 283. “were” should change to “was”
Response: Thanks very much for your great suggestion. We have revised it in our revised manuscript.
Line 286. “colony” should change to “colonies”
Response: Thanks very much for your great suggestion. We have revised it in our revised manuscript.
Line 307. “this” should change to “these”
Response: Thanks very much for your great suggestion. We have revised it in our revised manuscript.
Line 317. “was” should change to “were”
Response: Thanks very much for your great suggestion. We have revised it in our revised manuscript.
This manuscript is a resubmission of an earlier submission. The following is a list of the peer review reports and author responses from that submission.
Round 1
Reviewer 1 Report
In this ms. the authors used FISH and RT-PCR to study the following: A. The distribution and relative titer of CLas in various organs of D citri male and female adults of various ages post emergence; B. The distribution and relative titer of CLas in embryos and nymphs obtained from eggs of infected D. citri mothers. The first part (A) is mostly well excuted and well presented, but the second part (B) suffers from two main problems:
1. The authors did not indicate, in the Methods section (Line 279), which citrus plant they used for egg laying by infected females for studying transovarial transmission. If this plant was CLas-susceptible (e.g. the lemon used in other expts.) the resulting nymphs could have picked up CLas by feeding on plants recently infected by their mothers rather than directly through transovarial transmission. A resistant or immune citrus plant (e.g. Orange jasmine or Bergera koenigii) should have been used for egg laying in these expts. to avoid or reduce this possibilty. The authors should at least acknoledge this shortcoming and account for it in discussing their results.
2. The authors apparently assume that the detection of CLas in eggs and nymphs of infected mothers at high frequency, means automatically a high percentage of transovarial transmission, i.e. that these nymphs and resulting adults will transmit CLas effeciently to healthy plants. Thus, they confuse the term "infected" with "infective". It is well documentted that even with highly infected colonies of D. citri, when tested individually they will transmit this bacterium to only up to 20-46% of test plants in different world regions (reviewed by Ammar et al. 2020). As to transovarial transmission of CLas in D. citri, much lower percentages (2-4%) have been reported (Pelz-Stelinsky et al. 2010). Thus, I suggest that the authors stick to their findings without overinterpretation. They should indicate that a high percentage of emberyos and nymphs resulting from infected D. citri mothers were infected with CLas, and that actual transovrial transmission of this bacterium to healthy plants remains to be tested in future investigatiins.
Additionally, several mistakes in English, that made many sentences unclear throughout the text, should be revised and corrected.
Other detailed suggestions:
Title. Change " verified .... etc. " to "in different stages of embryos, nymphs and adults of Diaphorina citri".
Lines 13-14. Spell out CLas and D. citri before these abbreviations.
L. 25-26. Delete the sentence starting with " This is the first ... etc.".
L. 164. Delete "Verification of".
L. 178. In this Figure, and in Fig. S3, it is indicated that the scale bar is 100 not 200 um. Which is correct?!.
L. 190-191. Sentence needs correction to clarify the meaning.
L. 242. After "females" add "to their progeny".
L. 242-243. Delete this sentence, and replace it with "In future studies, actual transovarial transmission of CLas from infected D. citri mothers to their offspring needs to be verified by feeding first generation nymphs, individually or in small groups, on young susceptible healthy citrus plants. These tests could show if the high percentage of transovrially infected nymphs would lead to efficient inoculation of CLas into these test plants, and the potential significance of transovarial transmission in the epidemiology of CLas/HLB."
Methods.
L. 273. How many insects were included in each "biological replicate" ?!
L. 279. Which healthy citrus plant genotype or var. was used for egg laying? Very important as indicated above.
L. 287. When processing intact eggs or nymphs for FISH, did the authors use any cuts or chemicals to facilitate penetration of various reagents through the chorion or cuticle?
Author Response
Dear Editors and Reviewers:
Thank you for your letter and for the reviewers’ comments concerning our manuscript entitled “Spatial distribution and temporal dynamics of Candidatus Liberibacter asiaticus in different stages of embryos, nymphs and adults of Diaphorina citri” (Manuscript ID: ijms-2288434). These comments are all of great important and very helpful for revising and improving our manuscript. We have revised the manuscript carefully according to all your comments. Revised portions are marked with red font in our revised manuscript. Our point-by-point responses to the comments are listed below.
Sincerely,
Xiaoge Nian
Editors
Before being considered for publication, please rewrite the material and method section, adding more precision to the transovarial passage, as suggested by reviewer 2. Also, consider adding negative control in your figures. Finally, consider the scientific sound remarks of reviewer 1, point 2 .
Response: Thanks for your suggestion. We have revised the manuscript carefully according to all your comments. Revised portions are marked with red font in our revised manuscript.
Reviewer: 1
In this ms. the authors used FISH and RT-PCR to study the following: A. The distribution and relative titer of CLas in various organs of D citri male and female adults of various ages post emergence; B. The distribution and relative titer of CLas in embryos and nymphs obtained from eggs of infected D. citri mothers. The first part (A) is mostly well excuted and well presented, but the second part (B) suffers from two main problems:
1. The authors did not indicate, in the Methods section (Line 279), which citrus plant they used for egg laying by infected females for studying transovarial transmission. If this plant was CLas-susceptible (e.g. the lemon used in other expts.) the resulting nymphs could have picked up CLas by feeding on plants recently infected by their mothers rather than directly through transovarial transmission. A resistant or immune citrus plant (e.g. Orange jasmine or Bergera koenigii) should have been used for egg laying in these expts. to avoid or reduce this possibilty. The authors should at least acknoledge this shortcoming and account for it in discussing their results.
Response: Thanks for your suggestion. Kelley and Pelz-Stelinski (2019) reported that nymphs exposed to CLas via horizontal transmission of the oviposition site were more likely to pick up CLas rather than from veritical transmission. In order to reduce this possibility, CLas partially resistant citrus plants - orange Jasmine (Murraya panifulata (L.) Jack) (Alves et al., 2021) were used for egg laying by infected females for studying transovarial transmission. We have revised this sentence to the female adults reared on the CLas-positive plants were allowed to lay eggs on healthy orange Jasmine (M. panifulata (L.) Jack) plants. ” in Line 373-374.
2. The authors apparently assume that the detection of CLas in eggs and nymphs of infected mothers at high frequency, means automatically a high percentage of transovarial transmission, i.e. that these nymphs and resulting adults will transmit CLas effeciently to healthy plants. Thus, they confuse the term "infected" with "infective". It is well documentted that even with highly infected colonies of D. citri, when tested individually they will transmit this bacterium to only up to 20-46% of test plants in different world regions (reviewed by Ammar et al. 2020). As to transovarial transmission of CLas in D. citri, much lower percentages (2-4%) have been reported (Pelz-Stelinsky et al. 2010). Thus, I suggest that the authors stick to their findings without overinterpretation. They should indicate that a high percentage of emberyos and nymphs resulting from infected D. citri mothers were infected with CLas, and that actual transovrial transmission of this bacterium to healthy plants remains to be tested in future investigatiins.
Response: Thanks very much for your great suggestion. We have revised them and added in the title, abstract, discussion section in our new revised manuscript.
Additionally, several mistakes in English, that made many sentences unclear throughout the text, should be revised and corrected.
Other detailed suggestions:
Title. Change " verified .... etc. " to "in different stages of embryos, nymphs and adults of Diaphorina citri".
Response: Thanks very much for your great suggestion. We have revised the title “Spatial distribution and temporal dynamics of Candidatus Liberibacter asiaticus in different stages of embryos, nymphs and adults of Diaphorina citri” in our new revised manuscript.
Lines 13-14. Spell out CLas and D. citri before these abbreviations.
Response: Thanks very much for your great suggestion. We have revised them in Line 15-16.
L. 25-26. Delete the sentence starting with " This is the first ... etc.".
Response: Thanks very much for your great suggestion. We have deleted it in Line 28.
L. 164. Delete "Verification of".
Response: Thanks very much for your great suggestion. We have deleted in Line 212.
L. 178. In this Figure, and in Fig. S3, it is indicated that the scale bar is 100 not 200 um. Which is correct?!.
Response: Thanks very much for your great suggestion. The scale bar is 100 not 200 um. We have revised it in the Figure 7 and Figure S3 legends.
L. 190-191. Sentence needs correction to clarify the meaning.
Response: Thanks very much for your great suggestion. We have revised this sentence to “Therefore, combining the results of FISH and qRT-PCR, the difference of CLas in D. citri can be quantified comprehensively and accurately” in 252-254.
L. 242. After "females" add "to their progeny".
Response: Thanks very much for your great suggestion. We have added it in Line 309.
- 242-243. Delete this sentence, and replace it with "In future studies, actual transovarial transmission of CLas from infected D. citri mothers to their offspring needs to be verified by feeding first generation nymphs, individually or in small groups, on young susceptible healthy citrus plants. These tests could show if the high percentage of transovrially infected nymphs would lead to efficient inoculation of CLas into these test plants, and the potential significance of transovarial transmission in the epidemiology of CLas/HLB."
Response: Thanks very much for your great suggestion. We have deleted and added these sentence to “Kelley and Pelz-Stelinski (2019) reported that nymphs exposed to CLas via horizontal transmission of the oviposition site were more likely to pick up CLas rather than from vertical transmission [18]. In order to reduce this possibility, CLas partially resistant citrus plants - orange Jasmine (Murraya panifulata (L.) Jack) [31] were used for egg laying by infected females for studying transovarial transmission. CLas was observed in all laid eggs and subsequent 1st-2nd instar nymphs, indicating that a high percentage of emberyos and nymphs resulting from infected D. citri mothers were infected with CLas. Although in M. panifulata, CLas decreased gradually to undetectable level, it is also possible that the older nymphs may acquire the pathogen from the oviposition site. In addition, the epidermis of the nymph thickens gradually along with the development of the nymph, which is difficult for the nymphs to sufficiently decolorize, so CLas probe could not penetrate the cuticle in the 3rd to 5th instar nymphs. Hence, the distribution and relative titers of CLas in subsequent 3rd-5th instar nymphs were not showed in the present work. It is well documented that even with highly infected colonies of D. citri, when tested individually they will transmit this bacterium to only up to 20-46% of test plants in different world regions [32]. As to transovarial transmission of CLas in D. citri, much lower percentages have been reported [17]. In future studies, actual transovarial transmission of CLas from infected D. citri mothers to their offspring needs to be verified by feeding first generation nymphs, individually or in small groups, on young susceptible healthy citrus plants. These tests could show if the high percentage of transovarially infected nymphs would lead to efficient inoculation of CLas into these test plants, and the potential significance of transovarial transmission in the epidemiology of CLas/HLB."
in Line 309-331.
Methods.
L. 273. How many insects were included in each "biological replicate" ?!
Thanks very much for your question. “Each replication included 40-45 insects for qRT-PCR. Four FISH tests were conducted and a total of 60 sample per tissue were imaged to confirm the repeatability”, and we have added these in Line 366-367.
L. 279. Which healthy citrus plant genotype or var. was used for egg laying? Very important as indicated above.
Thanks very much for your question. CLas partially resistant citrus plants - orange Jasmine (Murraya panifulata (L.) Jack) (Alves et al., 2021) were used for egg laying by infected females for studying transovarial transmission. We have revised this sentence to the female adults reared on the CLas-positive plants were allowed to lay eggs on healthy orange Jasmine (M. panifulata (L.) Jack) plants. ” in Line 373-374.
L. 287. When processing intact eggs or nymphs for FISH, did the authors use any cuts or chemicals to facilitate penetration of various reagents through the chorion or cuticle?
Thanks very much for your question. As the procedure described by FISH, before added CLas probe, the samples were washed with 100% ethanol for three times (5 min/ each time), and sufficiently decolorized with 6% H2O2 in 80% ethanol.
Reviewer 2 Report
Nian et al., did a study using FISH as the technique to assess organ infectivity of Candidatus Liberibacter asiaticus (CLas) in Diaphorina citri, the asian citrus psyllid (ACP) and vertical transmission to offspring population.
Ammar et al. 2011 (10.1111/j.1439-0434.2011.01836.x) have shown by FISH that Las infect most of the ACP organs, including reproductive organs, event though at low rates. That was a further explanation to the low detection rates, previously found by Pelz-Stelinksi et al., 2010 (10.1603/EC10123) by using PCR. Besides, Pelz-Stelinksi et al., 2010 (10.1603/EC10123) reports low rates of transovarial transmission, that is, passage of Las from mother to offspring. Transmission experiments were carefully conducted, transferring eggs to hatch in Bergera koenigii plants, a known Las resistant plant (Beloti et al., 2018 10.1094/PHYTO-01-18-0012-R). B. koenigii was subsequently shown to be a non-host of Las in such a way that Las titer declines after ACP inoculation by Alves et al., 2021 (10.3389/fmicb.2021.683923). That is an absolute important aspect to be taken into account, since Lee et al., 2015 (10.1073/pnas.1508253112) was able to show asymptomatic Las acquisition by ACP, when hatching and developing as nymphs in newly Las infect citrus tissues by the mother.
In the current manuscript under evaluation, authors mention only briefly the “Sample Preparation” section. There is poor description of the material and methods concerning transmission of Las from mother to offspring.
Section 4, line 253 material and methods. Poor description of the assay and the controls to show the FISH results are indeed from the CLas. Even if CLas was indeed shown in the offspring, the experimental set up, laying eggs in citrus and allowing them develop into nymphs and adults in the same plant adults feed and eventually inoculate Las, allows the asymptomatic acquisition of Las from the plant the adult feed (Lee et al., 2015). There is no proof of transovarial transmission. Besides, material and section are not comprehensive enough.
At the result section, figures 1 to 6, there is no negative control? There is Las even in the wings (Figure 7)? Why not showing instars 3rd, 4th, and 5th? FISH technique was accurately set up for the experiment? It seems not. Please argue.
Author Response
Dear Editors and Reviewers:
Thank you for your letter and for the reviewers’ comments concerning our manuscript entitled “Spatial distribution and temporal dynamics of Candidatus Liberibacter asiaticus in different stages of embryos, nymphs and adults of Diaphorina citri” (Manuscript ID: ijms-2288434). These comments are all of great important and very helpful for revising and improving our manuscript. We have revised the manuscript carefully according to all your comments. Revised portions are marked with red font in our revised manuscript. Our point-by-point responses to the comments are listed below.
Sincerely,
Xiaoge Nian
Editors
Before being considered for publication, please rewrite the material and method section, adding more precision to the transovarial passage, as suggested by reviewer 2. Also, consider adding negative control in your figures. Finally, consider the scientific sound remarks of reviewer 1, point 2 .
Response: Thanks for your suggestion. We have revised the manuscript carefully according to all your comments. Revised portions are marked with red font in our revised manuscript.
Reviewer: 2
Nian et al., did a study using FISH as the technique to assess organ infectivity of Candidatus Liberibacter asiaticus (CLas) in Diaphorina citri, the asian citrus psyllid (ACP) and vertical transmission to offspring population.
Ammar et al. 2011 (10.1111/j.1439-0434.2011.01836.x) have shown by FISH that Las infect most of the ACP organs, including reproductive organs, event though at low rates. That was a further explanation to the low detection rates, previously found by Pelz-Stelinksi et al., 2010 (10.1603/EC10123) by using PCR. Besides, Pelz-Stelinksi et al., 2010 (10.1603/EC10123) reports low rates of transovarial transmission, that is, passage of Las from mother to offspring. Transmission experiments were carefully conducted, transferring eggs to hatch in Bergera koenigii plants, a known Las resistant plant (Beloti et al., 2018 10.1094/PHYTO-01-18-0012-R). B. koenigii was subsequently shown to be a non-host of Las in such a way that Las titer declines after ACP inoculation by Alves et al., 2021 (10.3389/fmicb.2021.683923). That is an absolute important aspect to be taken into account, since Lee et al., 2015 (10.1073/pnas.1508253112) was able to show asymptomatic Las acquisition by ACP, when hatching and developing as nymphs in newly Las infect citrus tissues by the mother.
In the current manuscript under evaluation, authors mention only briefly the “Sample Preparation” section. There is poor description of the material and methods concerning transmission of Las from mother to offspring.
Response: Thanks very much for your great suggestion. Based on the reviewers and editor suggestion, we have revised and added information as following:
- Revise the title to “Spatial distribution and temporal dynamics of Candidatus Liberibacter asiaticus in different stages of embryos, nymphs and adults of Diaphorina citri”.
- Revise 3subtitle to “4.3 Preparation of embryos and subsequent nymphs” in Line 370;
- Add the information“In order to explor the dynamics of CLas in different stages of embryos and subsequent nymphs of citri, the female adults reared on the CLas-positive plants were allowed to lay eggs on healthy orange Jasmine (M. panifulata (L.) Jack) plants” in Line 372-374;
- Add “In order to explore the dynamics of CLas in different stages of embryos and subsequent nymphs of D. citri, healthy orange Jasmine ( panifulata(L.) Jack) plants were used for egg laying by infected females for studying the dynamics of CLas in different stages of embryos and subsequent nymphs of D. citri” in Line 351-354.
- Add “All the experiments were performed with four biological replications. Each replication included 40-45 insects for qRT-PCR. Four FISH tests were conducted and a total of 60 sample per tissue were imaged to confirm the repeatability”in Line 367-369.
- Add “The adult females and males at 15 DAE of citrireared on CLas-negative C. limon plants were dissected for negative control” in Line 371-373.
- Add “3 Preparation of embryos and subsequent nymphs. In order to explore the dynamics of CLas in different stages of embryos and subsequent nymphs of D. citri, the female adults reared on the CLas-positive plants were allowed to lay eggs on healthy orange Jasmine (M. panifulata(L.) Jack) plants” in Line 375-379.
- Add “Four biological repetition were conducted and imaged to confirm repeatability. As for FISH, a total of 200 embryos of each stage and 60 nymphs each instar were tested. As for qRT-PCR, the samples of different stage of embryos and nymphs were gently scrape off the plants with a soft brush, and each repetition included 40-50 nymphs”in Line 386-391.
Section 4, line 253 material and methods. Poor description of the assay and the controls to show the FISH results are indeed from the CLas. Even if CLas was indeed shown in the offspring, the experimental set up, laying eggs in citrus and allowing them develop into nymphs and adults in the same plant adults feed and eventually inoculate Las, allows the asymptomatic acquisition of Las from the plant the adult feed (Lee et al., 2015). There is no proof of transovarial transmission. Besides, material and section are not comprehensive enough.
Response: Thanks very much for your great suggestion. Based on the Reviewers and editor suggestion, we have revised and added information as following:
- Revise the title to “Spatial distribution and temporal dynamics of Candidatus Liberibacter asiaticus in different stages of embryos, nymphs and adults of Diaphorina citri”.
- Revise 3subtitle to “4.3 Preparation of embryos and subsequent nymphs” in Line 370;
Add the information “In order to explor the dynamics of CLas in different stages of embryos and subsequent nymphs of D. citri, the female adults reared on the CLas-positive plants were allowed to lay eggs on healthy orange Jasmine (M. panifulata (L.) Jack) plants” in Line 372-374;
- Add “Kelley and Pelz-Stelinski (2019) reported that nymphs exposed to CLas via horizontal transmission of the oviposition site were more likely to pick up CLas rather than from vertical transmission [18]. In order to reduce this possibility, CLas partially resistant citrus plants - orange Jasmine (Murraya panifulata(L.) Jack) [31] were used for egg laying by infected females for studying transovarial transmission. CLas was observed in all laid eggs and subsequent 1st-2nd instar nymphs, indicating that a high percentage of emberyos and nymphs resulting from infected citri mothers were infected with CLas. Although in M. panifulata, CLas decreased gradually to undetectable level, it is also possible that the older nymphs may acquire the pathogen from the oviposition site. In addition, the epidermis of the nymph thickens gradually along with the development of the nymph, which is difficult for the nymphs to sufficiently decolorize, so CLas probe could not penetrate the cuticle in the 3rd to 5th instar nymphs. Hence, the distribution and relative titers of CLas in subsequent 3rd-5th instar nymphs were not showed in the present work. It is well documented that even with highly infected colonies of D. citri, when tested individually they will transmit this bacterium to only up to 20-46% of test plants in different world regions [32]. As to transovarial transmission of CLas in D. citri, much lower percentages have been reported [17]. In future studies, actual transovarial transmission of CLas from infected D. citri mothers to their offspring needs to be verified by feeding first generation nymphs, individually or in small groups, on young susceptible healthy citrus plants. These tests could show if the high percentage of transovarially infected nymphs would lead to efficient inoculation of CLas into these test plants, and the potential significance of transovarial transmission in the epidemiology of CLas/HLB."
in Line 309-331.
At the result section, figures 1 to 6, there is no negative control? There is Las even in the wings (Figure 7)? Why not showing instars 3rd, 4th, and 5th? FISH technique was accurately set up for the experiment? It seems not. Please argue.
Question 1: Negative control.
Response 1: Thanks very much for your great suggestion. The adult females and males at 15 DAE of D. citri reared on CLas-negative C. limon plants were dissected for negative control. We have added the negative control in the figure S1 and S3 in the supplemental data.
Question 2: Why not showing instars 3rd-5th.
Response 2: Thanks very much for your great suggestion. There are two reasons for us. On the one hand, Kelley and Pelz-Stelinski (2019) reported that nymphs exposed to CLas via horizontal transmission of the oviposition site were more likely to pick up CLas rather than from vertical transmission [18]. In order to reduce this possibility, CLas partially resistant citrus plants - orange Jasmine (Murraya panifulata (L.) Jack) [31] were used for egg laying by infected females for studying the dynamics of CLas in different stages of embryos and subsequent nymphs of D. citri. Although in M. panifulata, CLas decreased gradually to undetectable level, it is also possible that the older nymphs may acquire the pathogen from the oviposition site. On the other hand, the epidermis of the nymph thickens gradually along with the development of the nymph, which is difficult for the nymphs to sufficiently decolorize, so CLas probe could not penetrate the cuticle in the 3rd to 5th instar nymphs. Hence, the distribution and relative titers of CLas in subsequent 3rd-5th instar nymphs were not showed in the present work.
Question 3: Whether there is Las signal even in the wings.
Response 3: Thanks very much for your great suggestion. We checked all the FISH results about 2nd instar nymphs. There is indeed Las signal even in the wings. The wing buds were began to develop on 2nd instar nymphs, which was different from adult wings. In addition, in our negative control, we did not see CLas signal in the wings.